# Inhibition of Arp2/3 Complex after ADP-Ribosylation of Arp2 by Binary *Clostridioides* Toxins

**DOI:** 10.3390/cells11223661

**Published:** 2022-11-18

**Authors:** Carsten Schwan, Alexander E. Lang, Andreas Schlosser, Setsuko Fujita-Becker, Abdulatif AlHaj, Rasmus R. Schröder, Jan Faix, Klaus Aktories, Hans Georg Mannherz

**Affiliations:** 1Institute of Experimental and Clinical Pharmacology and Toxicology, Faculty of Medicine, Albert-Ludwig-University, 79104 Freiburg, Germany; 2Rudolf Virchow Center of Experimental Biomedicine, University of Würzburg, 97080 Würzburg, Germany; 3Cryo-Electron Microscopy, BioQuant, University Hospital, 69120 Heidelberg, Germany; 4Department of Anatomy and Molecular Embryology, Ruhr-University, 44780 Bochum, Germany; 5Department of Cellular Physiology, Ruhr-University, 44780 Bochum, Germany; 6Institute of Biophysical Chemistry, Hannover Medical School, 30625 Hannover, Germany; 7Department of Anatomy and Molecular Embryology and of Cellular Physiology, Ruhr-University, Universitätsstr. 150, 44780 Bochum, Germany

**Keywords:** actin, ADP-ribosyltransferases, Arp2/3 complex, *Clostridioides* binary toxins

## Abstract

Clostridioides bacteria are responsible for life threatening infections. Here, we show that in addition to actin, the binary toxins CDT, C2I, and Iota from *Clostridioides difficile*, *botulinum*, and *perfrigens*, respectively, ADP-ribosylate the actin-related protein Arp2 of Arp2/3 complex and its additional components ArpC1, ArpC2, and ArpC4/5. The Arp2/3 complex is composed of seven subunits and stimulates the formation of branched actin filament networks. This activity is inhibited after ADP-ribosylation of Arp2. Translocation of the ADP-ribosyltransferase component of CDT toxin into human colon carcinoma Caco2 cells led to ADP-ribosylation of cellular Arp2 and actin followed by a collapse of the lamellipodial extensions and F-actin network. Exposure of isolated mouse colon pieces to CDT toxin induced the dissolution of the enterocytes leading to luminal aggregation of cellular debris and the collapse of the mucosal organization. Thus, we identify the Arp2/3 complex as hitherto unknown target of clostridial ADP-ribosyltransferases.

## 1. Introduction

Anaerobic clostridial bacteria are causative for a number of severe human infectious diseases. A number of their proteinaceous toxins are crucial virulence factors of these pathogens. In many cases the cytoskeleton of host cells is the preferred target of these virulence factors. An important group are the toxins CDT from *Clostridioides difficile*, C2I from *Clostridioides botulinum* and Iota from *Clostridioides perfringens* [1,2,3]. These toxins are binary in structure and consist of an enzyme component with ADP-ribosyltransferase (ART) activity and a separate membrane binding component, which is responsible for cellular toxin binding and uptake [4]. The membrane binding component attaches to a specific receptor, oligomerizes, and binds the ART containing component. Subsequently, this complex containing both toxin components is endocytosed. After acidification of the endosome the membrane binding component forms a pore allowing release of the ART containing component through the endosomal membrane into the cytosol (for a review see [5]). There the ART containing toxins ADP-ribosylate actin at Arg177 leading to inhibition of its polymerizability and the partial disassembly of intracellular actin filaments ([3]; reviewed in [1,2]).

Recently, hypervirulent and multidrug resistant strains of *C. difficile* were recognized, which during antibiotic treatment can multiply and colonize the colon causing severe diarrhea and pseudomembranous colitis accompanied by high morbidity and mortality [5,6,7]. These *C. difficile* strains are a main cause of hospitalization associated infections [7,8]. In addition to the prototypical Rho/Ras-glycosylating toxins TcdA and TcdB they produce the binary *C. difficile* toxin CDT composed of the ART containing CDTa and membrane binding component CDTb [6,8,9,10,11].

However, the precise pathogenic mechanisms of clostridial ADP-ribosylating toxins are still not completely understood. Therefore, we searched for additional substrate proteins of the bacterial ADP-ribosyltransferases. To this aim we analyzed ART-activity containing toxins including the *C. difficile* toxin CDT, C2I toxin from *C. botulinum*, and Iota toxin from *C. perfringens* for their ability to ADP-ribosylate the actin-related proteins of the Arp2/3 complex. The ubiquitous Arp2/3 complex consists of seven subunits, resides underneath the plasma membrane and is responsible for the stabilization of the cortical F-actin beneath the plasma membrane and its attachment to cell junctions. Furthermore, a number of stimulating signaling pathways converge on Arp2/3 complex, which subsequently regulates the formation of branched actin-filament networks within lamellipodia of migratory eukaryotic cells [12,13].

Our biochemical data showed that the actin-related protein 2 (Arp2) of the Arp2/3 complex is ADP-ribosylated by bacterial ADP-ribosyltransferases, in particular by the binary ART-activity containing CDT toxin of *C. difficile* toxin. In order to bridge the gap between protein modification by pathogens and possible clinical consequences, we subsequently infected human colon derived Caco2 cells and excised mouse colon pieces with this toxin to analyze its effects under conditions more closely resembling an in vivo situation. The results obtained from these experiments confirmed the ADP-ribosylation of Arp2 and at the same time that of actin at an apparently equal amount. Obviously, their modifications resulted in dramatic alterations in cell and tissue morphology suggesting that the ART-activity containing CDT toxin of *C. difficile* might suffice to provoke the pathological changes leading to the typical severe colitis.

## 2. Materials and Methods

### 2.1. Materials

Fetal calf serum (FCS) and media were obtained from Gibco (Deisenhofen/Germany). The monoclonal anti-actin antibody (clone AC74) was purchased from Sigma-Aldrich (Poole, Dorset, UK) and FITC-labelled anti-rabbit IgG from Amersham (Amersham Life Science/UK). TRITC-phalloidin was obtained from Molecular Probes (Eugene, OR, USA). The fluorescent etheno-NAD was obtained from Sigma-Aldrich (Munich, Germany)

### 2.2. Protein Expression and Analysis

Rabbit skeletal muscle actin was prepared from dried acetone powder obtained from fresh rabbit psoas muscle as described [14]. G-actin was stored in G-buffer (5 mM HEPES-OH, pH 7.4, 0.1 mM CaCl_2_, 0.5 mM NaN_3_, and 0.2 mM ATP, pH 7.4). The Arp2/3 complex was purified from *Acanthamoeba castellani* and its mammalian variant from pig brain as described [15,16]. The activating C-terminal VCA domain of WASp (containing Verprolin-like, Central and Acidic regions) from N-WASP (comprising residues 392 to 501) was expressed in *Escherichia coli* and purified according to [15]; both were kindly supplied by Prof. D.L. Barber and Dr. A. Schoenichen (University of California, San Francisco, CA, USA). The clostridial toxins (CDT, C2I and Iota-a toxin from *Clostridioides difficile, botulinum and perfrigens*, respectively) with ADP-ribosyl-transferase (ART) activities were prepared recombinantly as described previously [17,18]. The components a and b of the CDT toxin from *Clostridioides difficile* (CDTa containing the ART and CDTb the membrane translocator activity) were expressed recombinantly in *E. coli* and due to their N-terminal His-tag purified by Ni-NTA affinity binding [17]. The TccC3 toxin from *Photorhabdus luminescense* was prepared as detailed [19] and the FH2 segments of the formins mDia1 (comprising residues 826–1163) and mDia3 (comprising residues 701–1061) were expressed in *E. coli* and purified as described [20] and kindly supplied by Prof. A. Wittinghofer (Max-Planck-Institute of Molecular Physiology, Dortmund, Germany).

### 2.3. Cell Culture and Intoxication and Immunostaining

The human colon carcinoma cell line Caco2 was obtained from CLS (Cell Lines Service, Germany, independent cell repository). The cells are available from ATCC (ATCC number: HTB-37; human origin with ethnicity Caucasian. Donor age 72 years; male and from colon). These epithelial-like cells were grown and maintained at 37 °C and 5% CO_2_ in plastic flasks in DMEM (Dulbecco modified Eagle medium) supplemented with 10% FCS. Since Caco2 cells express the CDTb receptor lipolysis-stimulated lipoprotein receptor (LSR) intoxication of the CDTa ART was achieved by simultaneous exposure of Caco2 cells grown on sterile glass coverslips to 500 ng/mL CDTb and 200 ng/mL CDTa under cell culture conditions as detailed previously [18]. After increasing incubation periods, the cells were fixed by addition of 4% paraformaldehyde for 20 min.

Colon pieces were taken from male wild type mice of the C57BL/6 genetic background. The mice were maintained in the local animal house of the Ruhr-University observing the regulations of the German Animal Protection Law and sacrificed by cervical dislocation following the recommendations of the Animal Care and Use Committee of the Ruhr-University, Bochum, in compliance with the German guidelines for animal care and procedures. The colon pieces were exposed to CDTa and CDTb, fixed by 4% paraformaldehyde and immunostained as further detailed below.

For immunostaining the cells and the colon pieces were permeabilized with 0.2% Triton X-100 in PBS for 5–10 min, washed three times in PBS, incubated with primary antibodies: mouse monoclonal anti-Arp2 (FMS96; Abcam, Cambridge, UK; kindly provided by Prof. S. Linder, Hamburg, Germany) or affinity-purified rabbit anti-ArpC1 (40 kDa subunit; Abcam) at dilutions of 1:50 or 1:100, respectively, at 4 °C overnight. Subsequently the slides were incubated with Alexa Fluor^®^-568-labelled phalloidin and FITC-labelled secondary antibodies (Sigma-Aldrich, Munich, Germany) for 1h at RT as detailed [6,9]. The nuclei were visualized with Hoechst 33342 (Riedel-de-Haen; Schwerte, Germany). Finally, the coverslips were mounted with Dako Cytomatic fluorescent mounting medium. Immunocytochemical stainings were analyzed using a Zeiss LSM 800 confocal laser scanning microscope.

### 2.4. Analytical Procedures

Protein concentrations were determined by the colorimetric assay [21]. SDS-PAGE was performed using 7.5% or 10% (*w*/*v*) polyacrylamide gels unless stated otherwise. Trichloroethanol at 0.5% was included in the separation gel to fluorescently visualize the separated protein bands before Western blotting.

For immune-precipitation of Arp2/3 complex, cell or tissue homogenates were prepared in RIPA-buffer (250 mM NaCl, 5 mM EDTA, 50 mM NaF, 1% NP-40, 0.1% NaN_3_, and 50 mM Tris-HCl, pH 7.4) and frozen at −20 °C until use. After two cycles of freezing and thawing, the samples were centrifuged at 14,000 rpm for 30 min. by an Eppendorf bench centrifuge and the supernatants were collected. Then, about 50 µg (in about 100 µL) of the respective supernatant was supplemented with 2 µL of the monoclonal anti-Arp2 antibody (FMS96; Abcam, Cambridge, UK) and incubated for 1 h at 25 °C. Subsequently, 20 µL of Protein A insolubilized to Sepharose 4B (Sigma, Munich, Germany) was added and after an incubation for 1 h at 25 °C centrifuged for 10 min at 14,000 rpm. The pellet was washed 5 times with 0.5 mL HEPES-buffer. The supernatants of each washing step were carefully removed and either discharged or stored for SDS-PAGE.

Western blots were performed as detailed previously [22]. Autoradiography was used to identify ADP-ribosylated components of the Arp2/3 complex by supplementing NAD with ^32^P-labelled NAD (obtained from Perkin Elmer, Rodgau, Germany) following the procedure described in [19,23].

For measuring the polymerization kinetics by the pyrene-assay, G-actin was modified at Cys374 by pyrenyl-iodoacetamide (pyrene-actin) as described [24]. Pyrene-labelled actin was added to 5% of the total actin concentration. Polymerization was initiated by addition of 2 mM MgCl_2_ and 50 mM KCL and the increase of pyrene-actin fluorescence was determined using a Shimadzu RF-5001-PC spectrofluorometer at wavelength settings for excitation and emission of 365 nm and 385 nm, respectively.

### 2.5. Electron Microscopy

Protein samples were diluted to 0.1 mg/mL in HEPES-buffer, pH 7.4, with 2 mM MgCl_2_ to trigger polymerization. For negative staining, 4 µL of each sample were adsorbed to freshly glow-discharged carbon-coated copper grids (200 mesh) for 45 sec. After washing with buffer, the grids were incubated for 45 s on a drop of 1% uranylacetate [24]. Excess staining solution was removed with filter paper and then the grids were air-dried. Three different grids were prepared for each sample. Digital micrographs were then recorded with a Zeiss transmission electron microscope EM923 run at 120 kV fitted with a TemCamF416 camera (Tietz Video and Image Processing Systems, Gauting, Germany). The number of filaments longer than 0.1 micrometer and the number of branches were counted manually on the EM micrographs.

### 2.6. Determination of Filament Branching by Fluorescence Microscopy

Freshly purified skeletal muscle actin at 4.8 µM (0.2 mg/mL) was mixed with 50 nM native or ADP-ribosylated Arp2/3 complex from *Acanthaemoeba castellani* or pig brain. After addition of 0.1 mg/mL VCA peptide it was polymerized by 10 mM HEPES-HCl buffer, pH 7.4 (containing 50 mM KCl, 2 mM MgCl_2_, 0.1 mM CaCl_2_, and 0.2 mM ATP; buffer A). After dilution to 1 µM and the F-actin samples were stained with 2 µM TRITC-phalloidin (SIGMA, Munich, Germany) and further incubated for 60 min. Then, the TRITC-phalloidin stained actins were diluted to 10 nM in buffer A and 3 μL were placed on a glass-slide, mixed with 3 μL of DAKO fluorescence mounting medium (Agilent DAKO, Santa Clara, CA, USA/Glostrup, Denmark) and covered with a coverslip.

Subsequent fluorescence microscopy was performed using Zeiss AxioImager Z2m microscope equipped with a Zeiss LD LCI Plan-Apochromat 63×/1.2 multi-immersion objective and Zeiss Axiocam 503 color camera. Glycerol was used as immersion medium. Rhodamine fluorescence was excited using the LED 555 of the solid-state light source Colibri 7 and the quadruple bandpass filter set 90 HE, both from Zeiss. Images were recorded as gray-scale pictures with the microscope-associated ZEN software. The image size was 1936 × 1460 pixels, and the pixel size was 0.116 µm/pixel. Due to prolonged snap time and slight drift of the sample double images were collected which became visible only at higher magnifications but allowed to unequivocally differentiate between filament branching or crossing.

### 2.7. Analytical Tools

Recorded images from fluorescence microscopy were analyzed to obtain the number of the filaments and junction points, using ImageJ and the available plugin Ridge Detection (URL: https://imageJ.net/plugins/ridge-detection, accessed on 5 August 2022). This plugin is based on the detection algorithm described by Steger [25] for detecting ridges and lines. The parameter was selected to indicate all visible filaments above 1 micrometer and their junction points. The final setting was line width: 4.0, sigma: 1.65, lower threshold: 2.72, upper threshold: 7.31, minimum line length: 8.60 (=1µm), and maximum line length was not defined. For some images with weak signal the upper threshold was reduced to 4.5. The locations of the indicated junction points were checked manually by zooming in on each one on the original image to determine whether they were branches, crossings or simply two filaments approaching each other, and only branches were counted. The number of identified branches and the number of filaments obtained by Ridge Detection were transcribed to Microsoft Excel [26], and we determined for each image the frequency of branching relative to the total number of filaments. Only the results of the analysis of images with maximally 300 filaments per image were used, as higher filament density increased the frequency of filament overlap rendering it difficult to distinguish between crossing and branching.

### 2.8. Mass-Spectrometric (MS) Analysis

For in-gel digestion the excised gel bands were destained with 30% acetonitrile, shrunk with 100% acetonitrile, and dried in a Vacuum Concentrator (Concentrator 5301, Eppendorf, Hamburg, Germany). Digestions with trypsin (trypsin gold, mass spectrometry grade; Promega, Walldorf, Germany) were performed overnight at 37 °C in 0.05 M NH_4_HCO_3_ (pH 8). About 0.1 µg of protease was used for one gel band. Peptides were extracted from the gel slices with 5% formic acid. All LC-MS/MS analyses were performed with the 1200 Agilent Chip-HPLC system, either coupled to a Q-TOF (Agilent 6520) or an ion trap (Agilent 6340) mass spectrometer. Peptides were separated on an HPLC-Chip with an analytical column of 75-µm i.d. and 150 mm length and a 40-nL trap column, both packed with Zorbax 300SB C-18 (5 µm particle size). Peptides were eluted with a linear acetonitrile gradient with 1%/min at a flow rate of 300 nL/min (starting with 3% acetonitrile). The Q-TOF was operated in the 2 Ghz extended dynamic range mode. MS/MS analyses were performed using data-dependent acquisition mode. After a MS scan (2 spectra/s), a maximum of three peptides were selected for MS/MS (2 spectra/s). Singly charged precursor ions were excluded from selection. Internal calibration was applied using one reference mass.

ETD analyses on the ion trap were performed using data-dependent acquisition mode. After a MS scan (standard enhanced mode), a maximum of three peptides were selected for ETD-MS/MS (standard enhanced mode). The automated gain control (ICC) for MS scans was set to 350,000. The maximum accumulation time was set to 300 ms. The following ETD parameters were used. ICC target: 400,000, reaction time: 100 ms, cut-off: 140, resonance excitation (Smart Decomp) was used for doubly charged peptides.

Mascot Distiller 2.3 was used for raw data processing and for generating peak lists, essentially with standard settings for the Agilent Q-Tof and ion trap. Mascot Server 2.3 was used for database searching with the following parameters: peptide mass tolerance: 20 ppm (Q-Tof) 1.1 Da (ion trap), MS/MS mass tolerance: 0.05 Da (Q-Tof), 0.3 Da (ion trap), enzyme: “trypsin” with 2 uncleaved sites allowed for trypsin, variable modifications: Carbamidomethyl), Gln-pyroGlu (N-term. Q), and oxidation (M), ADP-ribosylat (R). For protein and peptide identification a small custom database containing the protein sequence of Arp2 was used. All MS/MS spectra identified as ADP-ribosylated were validated by manual spectra interpretation. Three peptides were identified containing ADP-ribosylated arginine residues of which Arg179 was most prominently modified (Figure 1; for details see also Figure 3).

### 2.9. Mass spectrometric Identification of Peptides Unique for Human Arp2

Fluorescent gel band was excised, destained with 30% acetonitrile in 0.1 M NH_4_HCO_3_ (pH 8), shrunk with 100% acetonitrile, and dried in a vacuum concentrator (Concentrator 5301, Eppendorf, Germany). Digest was performed with 0.1 µg trypsin overnight at 37 °C in 0.1 M NH_4_HCO_3_ (pH 8). After removing the supernatant, peptides were extracted from the gel slice with 5% formic acid, and extracted peptides were pooled with the supernatant.

NanoLC-MS/MS analysis was performed on an Orbitrap Fusion (Thermo Scientific, Waltham, MA, USA) equipped with a PicoView Ion Source (New Objective, Frederik, MD, USA) and coupled to an EASY-nLC 1000 (Thermo Scientific). Peptides were loaded on capillary columns (PicoFrit, 30 cm × 150 µm ID, New Objective) self-packed with ReproSil-Pur 120 C18-AQ, 1.9 µm (Dr. Maisch) and separated with a 30 min linear gradient from 3% to 30% acetonitrile and 0.1% formic acid and a flow rate of 500 nL/min.

Both MS and MS/MS scans were acquired in the Orbitrap analyzer with a resolution of 60,000 for MS scans and 7500 for MS/MS scans. HCD fragmentation with 35% normalized collision energy was applied. A Top Speed data-dependent MS/MS method with a fixed cycle time of 3 s was used. Dynamic exclusion was applied with a repeat count of 1 and an exclusion duration of 30 s; singly charged precursors were excluded from selection. Minimum signal threshold for precursor selection was set to 50,000. Predictive AGC was used with AGC a target value of 2 × 10^5^ or MS scans and 5 × 10^4^ for MS/MS scans. EASY-IC was used for internal calibration.

MS data was analyzed with PEAKS Studio X+ (Bioinformatics Solutions Inc., Waterloo, ON, Canada). Raw data refinement was performed with the following settings: Merge Options: no merge, Precursor Options: corrected, Charge Options: 1–6, Filter Options: no filter, Process: true, Default: true, Associate Chimera: yes. De novo sequencing and database searching were performed with a Parent Mass Error Tolerance of 10 ppm. Fragment Mass Error Tolerance was set to 0.02 Da, and Enzyme was set to trypsin with a maximum of 3 missed cleavages allowed. The following variable modifications have been used: Oxidation (M), pyro-Glu from Q (N-term Q), acetylation (protein N-terminal). A maximum of 6 variable PTMs were allowed per peptide. Database searching was performed against the human reference proteome (proteome ID UP000005640). Database search result was filtered to 1% PSM-FDR and protein –10lgP > 20.

## 3. Results

### 3.1. ADP-Ribosylation of Components of Isolated Arp2/3 Complex

The Arp2/3 complex is composed of the actin-related proteins Arp2 and Arp3 and five additional subunits named ArpC1 to ArpC5 with decreasing molecular mass (see SDS-PAGE of Arp2/3 complex purified from *Acanthamoeba castellani*, Figure 1D). Figure 1 shows that the toxins’ enzyme components C2I, CDTa, and Iota-a catalyzed the incorporation of ^32^P-ADP-ribose from ^32^P-labeled NAD^+^ in a number of components of Arp2/3 complex purified from *Acanthamoeba castellani* in vitro (Figure 1A–C). The actin-related protein 2 (Arp2) was most strongly labelled by all clostridial ARTs (Figure 1A–C). In addition, the additional components ArpC1 (40 kDa), ArpC2 (34 kDa), and ArpC3/or C4 (about 20 kDa) appeared to be also modified. These components were strongly modified by C2I (Figure 1A), whereas CDTa and Iota-a toxin most prominently ADP-ribosylated Arp2 (Figure 1B,C). The band higher than Arp3 in Figure 1B represents the auto-ADP-ribosylated CDTa component, since it occurs also in the absence of Arp2/3 complex (Figure 1B). We tested also the possibility whether the bacterial ARTs modify other actin nucleators like formins, though of different molecular architecture. No ^32^P-ADP-ribose incorporation was detected for the FH2-domains of mDia1 and mDia3 (not shown). Conversely, the TccC3 toxin, an ART from *Photorhabdus luminescense* [19], did not significantly lead to ^32^P-ADP-ribose incorporation into any of the Arp2/3 complex components (Figure 1E).

To further test the specificity of the ADP-ribosylation, we investigated the effect of ADP-ribose, the Arp2/3-activating WASP-VCA peptide (Wiskott-Aldrich syndrome Verprolin C-terminal acidic peptide), and of F-actin on this reaction. ADP-ribose led to only a slight reduction of ^32^P-ADP-ribose incorporation excluding the possibility of an unspecific transfer of ADP-ribose to for instance a nucleotide-binding site by the ART activity (Figure 1A–C). Similarly, the VCA peptide inhibited only slightly the Arp2/3 modification (Figure 1F). In contrast, addition of F-actin significantly reduced the modification by CDTa and led to an almost complete inhibition of the ADP-ribosylation by Iota-a toxin (Figure 1F). Of note, the added F-actin itself was not ADP-ribosylated in agreement with previous reports ([1,4] see also Figure 1A,C).

Finally, we compared the rates of ADP-ribosylation of G-actin and Arp2 of the Arp2/3 complex. Surprisingly, G-actin was ADP-ribosylated by CDTa much faster than Arp2 (Figure 1H). These data suggest that Arp2 was less accessible within the intact Arp2/3 complex probably due to a close contact with Arp3 and/or other subunits of the Arp2/3 complex. Indeed, it has been reported that Arp2 and Arp3 form an F-actin-like contact [27,28], which will allow ADP-ribosylation of Arp2 though at much lower rate than G-actin. Thus, our data indicate that the ARTs of clostridial toxins target specifically the Arp2 isoform of actin-related proteins and in addition some of the actin-unrelated additional subunits of the *Acanthamoeba* Arp2/3 complex. In contrast, Arp3 does not possess an arginine at the corresponding position [29] and therefore was not ADP-ribosylated. Since the CDTa toxin catalyzed most prominently ^32^P-ADP-ribosylation of Arp2, we concentrated in further analyses on the functional consequences of modifying reaction of Arp2/3 complex by this toxin assuming that alterations of its function were primarily due to Arp2 ADP-ribosylation.

In addition, we employed Arp2/3 complex isolated from pig brain to verify the ADP-ribosylation of Arp2 of mammalian Arp2/3 complex using fluorescently labelled NAD (etheno-NAD) under identical conditions (due to the disassembly of the radioactive laboratory, a fluorescence technique had to be employed). The data shown in Figure 2 gives the ADP-ribosylation of the components of mammalian Arp2/3 complex under different conditions. Incubation of the mammalian Arp2/3 complex (shown in Figure 2A) with the toxins CDTa, C2I, and Iota-a and 0.4 mM etheno-NAD for two hours led to ADP-ribosylation Arp2 as indicated by the fluorescent bands (Figure 2B; lanes 2′to 4’). After extension of the incubation time to 4 h additional labelling of bands at the height of ArpC1 and ArpC2 by CDTa was observed (Figure 2C; lanes 5 and 6) that was not influenced by the presence of the VCA-peptide (Figure 2C; lane 6). Unfortunately, free etheno-NAD migrated slower than the Coomassie blue stain close to the positions of the ArpC3–5 subunits. Reducing the etheno-NAD to 0.1 mM made it possible to resolve the migration of the presumed ArpC3,4 subunits from free etheno-NAD and to demonstrate weak labelling of supposedly ArpC3,4 by CDTa, C2I, and Iota-a (arrow in Figure 2D; lanes 7–9). Furthermore, pig brain Arp2/3 complex ADP-ribosylated with 0.8 µM etheno-NAD for 5 h by CDTa was gel-filtrated over a column (1.0 cm × 5 cm) collecting first a 500 µL and subsequently 200 µL fractions, which were analyzed by SDS-PAGE (15% acrylamide). The data obtained clearly demonstrated strong fluorescent labelling of 44 and 40 bands presumably corresponding to Arp2 and ArpC2 (Figure 2E,F). Only weak labeling was observed for ArpC4,5 (as indicated by an arrow in Figure 2F, whereas free etheno-NAD started to appear in tube 8). Generally, the fluorescence signals of labelled Arp2/3 complex subunits appeared rather weak. It is possible that the modification of the adenine moiety of etheno-NAD led to a reduction of its affinity to these ARTs as previously observed for etheno-ATP binding to G-actin [30].

### 3.2. Mass Spectrometry Identifies Arg179 of Acanthamoeba Arp2 as Main Residue Modified

Employing mass spectrometry, the arginines 41, 179 and 382 of *Acanthamoeba* Arp2 [28] were identified to be ADP-ribosylated by CDTa (Figure 3A–C). Notably, the most prominently modified residue of Arp2 was Arg179 (Figure 3B), which in sequence alignments corresponds to Arg177 of classical actins (Figure 1I). Due to the high sequence identity between *Acanthamoeba* and mammalian Arp2 [29,31], we expect that also arginine179 of mammalian Arp2 will have been modified by the ARTs employed (identical sequence of residues 174–179 of mammalian and *Acanthamoeba* Arp2: LPHLTR**.** The ADP-ribosylated arginine is in bold).

### 3.3. Inhibition of the Nucleating Activity of ADP-Ribosylated Arp2/3 Complex

Polymerization assays using pyrene-labeled skeletal muscle actin showed that after ADP-ribosylation the Arp2/3 complex was unable to stimulate actin polymerization when determined in the absence or presence of the stimulating VCA-peptide (Figure 4A). Similarly, electron microscopy after negative staining demonstrated that modified Arp2/3 complex generated less actin filaments (Figure 4D,E) than control actin alone (Figure 4B) and branches (white arrows in Figure 4B–E; red arrows indicate filament crossings) than in the presence of native Arp2/3 complex (Figure 4C). Counting the number of filaments and branches of 27 arbitrarily selected, non-overlapping EM-micrographs of identical size and magnification resulted in 35 and 67 branches within 475 and 524 filaments for F-actin alone and plus native *Acanthamoeba* Ap2/3 complex, respectively. In the presence of ADP-ribosylated Arp2/3 we counted 30 branches within 304 filaments. Thus, in the presence of ADP-ribosylated Arp2/3, the number of branches counted was almost equal to F-actin alone (Table 1). We also tested the effect of native Ap2/3 on ADP-ribosylated and therefore polymerization inhibited actin and accordingly found only very few filaments but considerable amounts of non-polymerized, aggregated actin (Figure 4F) supporting the notion that our preparation of ADP-ribosylated Arp2/3 was free of ART activity and no ADP-ribosylation of actin had occurred during the incubation of native actin with ADP-ribosylated Arp2/3 (Figure 4A,D,E).

We also performed a similar quantitative analysis after staining the F-actin-Arp2/3 complex mixtures with TRITC-phalloidin. F-actin alone at 1 µM or incubated with native or ADP-ribosylated *Acanthamoeba castellani* or pig brain Arp2/3 complex (at 100:1 ratio) was stained with 2 µM TRITC-phalloidin and after a further incubation period of 60 min diluted to 10 nM F-actin and subsequently examined by fluorescence microscopy (see Material and Methods) assuming that phalloidin will prevent its depolymerization [32]. Representative images of all incubations are given in Figure 3G–K (white arrows indicate branching events; red arrows filament crossings). The quantitative evaluation was performed by using ImageJ Ridge Detection analysis [25] and demonstrated that under the conditions used Arp2/3 complex from both sources led to a doubling of the branching events, whereas their ADP-ribosylation led to a reduction of branching events to a value almost identical to F-actin alone (see also Table 1). Branching in the presence of F-actin alone was possibly due the formation of the anti-parallel (lower) dimer during the initial phase of polymerization [33].

### 3.4. Effect of Arp2 ADP-Ribosylation on the Actin Cytoskeleton of Human Caco2 Cells

In order to test the effect of CDT intoxication under more in vivo conditions, we first analyzed the alterations of Arp2 distribution (by using a monoclonal anti-Arp2 antibody) and the actin and tubulin cytoskeleton of human colon carcinoma Caco2 cells after intracellular translocation of recombinant CDTa by CDTb (Figure 5A–C). Under control conditions, the Arp2 immunoreactivity co-localized with cortical F-actin concentrating at cell–cell contacts and in regions of lamellipodial extensions (Figure 5A). In addition, the perinuclear region showed a punctate and circular anti-Arp2 staining probably attached to vesicular structures (Figure 5A). After intoxication for 45 min, most of the lamellipodial extensions had disappeared and Arp2 appeared to form aggregates or clusters along the cell periphery and within the cytoplasm (Figure 5B). After 90 min, Arp2 was almost evenly distributed within the cytoplasm occasionally forming small aggregates (Figure 5C). Z-stacks were used to count the clusters detected by Arp2 immunofluorescence. Their number (median value: 33/control cell) increased about threefold after 45 min but decreased threefold below control level after 90 min of toxin exposure (Figure 5H). These changes were observed in about 70% of the exposed Caco2 cells. In addition, the actin cytoskeleton appeared almost completely disassembled, most probably depolymerized, and therefore not stained by TRITC-phalloidin. In contrast, the tubulin network maintained its basic organization though single microtubules had a wavy appearance. In addition, the tubulin network formed long cytoplasmic filipodia-like extensions as reported previously [18] (Figure 5C). Thus, these data showed that the CDT-toxin led to complete disassembly of the actin cytoskeleton of the Caco2 cells.

### 3.5. Evidence for Arp2 ADP-Ribosylation in CTDa Toxin Exposed Human Caco2 Cells

In order to verify Arp2 ADP-ribosylation in Caco2 cells under these conditions, we exposed Caco2 cells after CDTa translocation by CDTb for 2 to 3 h to 0.15 mM Triton X-100 to allow etheno-NAD diffusion into these cells. Phase contrast images of control Caco2 cells demonstrated that addition of 0.15 mM Triton X-100 had no effect on their morphology and actin cytoskeleton (not shown), whereas in the presence of CDTa and CDTb the peripheral cells of the Caco2 cell clusters clearly showed the alterations as shown in Figure 5A–C. The cells were harvested, washed three times with PBS, and finally taken up in RIPA buffer and frozen at −20 °C. After thawing, immune-precipitation (IP) was performed using a monoclonal anti-Arp2 antibody (see Material and Methods). Homogenates of control (absence of toxin) and CDTa-treated cells and material obtained by the IP and of preceding washing steps was analyzed by SDS-PAGE (Figure 5D and E showing fluorescent bands as visualized under UV-light). The CDTa-treated Caco2 cell homogenate (Figure 5E, lane 2) contained two main fluorescent bands, which appeared to migrate at the position of mammalian Arp2 (about 48 kDa) and of actin (42 kDa). Surprisingly, the homogenate of the control cells showed also a very feeble staining of these two bands (Figure 5E, lane 1) possibly due to the presence of endogenous ADP-ribosyltransferase activity in these cells. The final IP sample of CDT-exposed cells contained only one main band of about 48 kDa (Figure 5E, lane 5) that migrated slightly slower than etheno-ADP labelled Arp2 control *Acanthamoeba* Arp2/3 complex (MW = 44 kDa; Figure 5E, lane 6) and labelled control actin (Figure 5E, lane 7). After immunoblotting of the IP gel (shown in Figure 5E) with anti-Arp2 mAb only the 48 kDa band was stained (Figure 5F, lane 5) suggesting that the immune-precipitated fluorescent 48 kDa band was indeed Arp2, but also indicating that the anti-Arp2 mAb used does not recognize *Acanthamoeba* Arp2. The homogenate of the etheno-NAD exposed cells revealed an almost equally strong fluorescent 42 kDa band (Figure 5E, lane 2), which migrated at the same molecular mass as labelled control actin (Figure 5E, lane 7). Its identity with actin was verified after stripping the blot shown in Figure 4F and subsequent immunoblotting with ant-actin (Figure 5G. lanes 1, 2, and 7). Apparently, actin was co-immune-precipitated with Arp2, since the anti-actin immunoblot indicated its presence also in lane 5 of Figure 5F.

Because the same anti-Arp2 antibody was used for IP and the subsequent immunoblotting, we additionally verified the identity of the IP precipitated fluorescent band obtained with human Arp2 by mass spectrometry. Therefore, in a separate experiment the fluorescent band obtained by IP (as shown in Figure 5E, lane 5) from treated Caco2 cell homogenates was carefully excised, digested with trypsin and analyzed by nanoLC-MS/MS. The data obtained clearly indicated the presence or two peptides (residues 52 to 63 and 274 to 281) unique for human Arp2 (Figure 6) supporting the notion that the fluorescent band obtained by IP was indeed human Arp2.

### 3.6. Evidence for Arp2 ADP-Ribosylation in CTDa Toxin Exposed Mouse Colon Epithelium

Similar to the toxin exposure of Caco2 cells, we treated isolated mouse colon pieces with the binary CDT toxin of *C. difficile*. After thoroughly washing 1 to 1.5 cm long pieces of mouse colon with PBS, they were filled with about 1 mL PBS containing CDTa and CDTb, 0.4 mM etheno-NAD, and 0.15 mM Triton X-100 to allow diffusion of the fluorescent NAD into the enterocytes. The filled colon pieces were tightly ligated at both ends and incubated for 3 h at 37 °C and 5% CO_2_ in cell culture medium. Thereafter, the ligations were removed and without further rinsing small slices were embedded in TissueTek and quickly frozen in liquid nitrogen for cryo-sectioning. Alternatively, small colon pieces were frozen directly in liquid nitrogen and stored at −20 °C for homogenate preparation. Cryo-sections (about 8 µm thick) were stained with haematoxylin-eosin (H.E.) or TRITC-phalloidin and Hoechst 33342 or immunostained with anti-Arp2. Though the colon pieces were maintained under artificial, cell culture conditions, the H.E. staining of control colon pieces after 3 h indicated a relatively well-preserved morphology with clearly distinct layers comprising the mucosa, Lamina muscularis mucosae, a narrow Tela submucosa, and Tunica muscularis (Figure 7B). After 3 h incubation we did not observe gross morphological alterations between colon pieces filled with only PBS (Figure 7A) and with PBS plus 1.5 mM Triton X-100 (Figure 7A´) when examining the H.E stained sections (see also Figure 7B´). Immunostaining consecutive sections of colon pieces incubated with PBS plus 1.5 mM Triton X-100 with anti-Arp2, TRITC-phalloidin plus Hoechst 33342 demonstrated also the preservation of the typical intestinal layering, i.e., the organization of the mucosal layer (M) into crypts lined by enterocytes sitting on the Lamina propria (lp) and muscularis mucosae (mm) and a narrow Tela submucosa (ts) followed by the outer Tunica muscularis (tm), as indicated in Figure 5B´. As expected TRITC-phalloidin strongly stained the smooth muscle cells of the Lamina muscularis mucosae and the Tunica muscularis (Figure 7A´´,B´), whereas anti-Arp2 strongly stained the Lamina propria (probably fibroblastic cells) and weakly the luminal (apical) face of the enterocytes lining the crypts (arrowheads in Figure 7B,B´).

After exposure of the colon pieces for 3 h to CDTa,b toxin and etheno-NAD, the basic colon morphology was preserved (Figure 7C,D). Since it was not possible to detect etheno-NAD fluorescence on colon sections, we again performed immunostaining with anti-Arp2 and TRITC-phalloidin. After 3 h CDTa,b treatment the enterocytes appeared completely dissolved, only short anti-Arp-2 positive rudiments of the supporting Lamina propria were detectable (arrows in Figure 7C,D). The Lamina muscularis mucosae was not clearly discernable, whereas the Tunica muscularis appeared intact (Figure 7C´´´,D´´´). The lumen was filled with cell debris presumably originating from necrotic enterocytes that was weakly stained by anti-Arp2 and TRITC-phalloidin, but strongly by Hoechst 33342 (Figure 7C´´,D´´). The data obviously indicated selective degradation of the enterocytes, whereas the Arp2 positive fibroblastic cells of the supporting Lamina propria appeared less affected. This difference might be due to the presence of the lipolysis-stimulated lipoprotein receptor only on the colon enterocytes, which has been identified as receptor for the pore forming clostridial CDTb toxin [10].

Colon homogenates were analyzed after IP with anti-Arp2 mAb and SDS-PAGE by fluorescence for etheno-ADP-ribosylated Arp2 and by immunoblotting with anti-Arp2 and -actin mAb. The data showed in the IP fractions fluorescent bands, in particular a band migrating at 48 kDa (Figure 7D; the height of mammalian Arp2). Its identity with Arp2 was again verified by immunoblotting with anti-Arp2 (Figure 7E). After blot stripping, immunostaining with anti-actin revealed for the lower fluorescent band (about 42 kDa) a positive reactivity (Figure 7F) indicating co-precipitation of actin with Arp2. Again, Arp2 and the endogenous actin appeared to be of almost equal fluorescence and immunostaining intensity (Figure 7D–F, for details see figure legend).

## 4. Discussion

Previously it has been shown that the Arp2/3 complex is targeted by a number of bacterial pathogens. For instance, *Listeria monocytogenes* bacteria express ActA—a bacterial surface protein that activates Arp2/3 complex and uses it for comet tail formation and their intra- and intercellular transport [34]. Our data show for the first time that clostridial toxins with ART-activity also target the Arp2/3 complex by ADP-ribosylating the Arp2 subunit and some accessory proteins of the Arp2/3 complex. Indeed, Arp2 possesses about 50% sequence identity and high structural homology to actin [28,31]. Therefore, Arp2 is a well-suited second substrate for bacterial ARTs, since it also possesses an arginine at their target site (Arg179), whereas this residue is a histidine in Arp3 [29].

This Arp2 modification affects the whole Arp2/3 complex and leads to inhibition of its stimulatory activity on actin polymerization and filament branching. Since it has been shown that during branch formation the first actin molecule of the growing daughter filament attaches to the barbed end area of Arp2, the ADP-ribosylation of Arp2 at Arg179 could block the attachment of this first actin subunit similar to the inhibition of actin subunit addition to the plus end of F-actin by the capping activity of ADP-ribosylated actin [35].

When comparing the time dependence of ADP-ribosylation of Arp2 of the Arp2/3 complex with that of G-actin, it was obvious that G-actin is much faster modified that Arp2 (Figure 1H). Within F-actin the Arg177 residue is located at the interstrand interface [36,37] and therefore is not accessible for the ARTs although they may bind to F-actin. Structural studies have shown that *C. perfrigens* Iota-a toxin binds to a large target area of monomeric (G-) actin covering subdomains 1,3, and 4 [38], of which subdomains 3 and 4 are not fully exposed in F-actin [38]. Therefore, the ARTs will bind to F-actin with reduced affinity, but be unable to ADP-ribosylate Arg177 of actin, thus explaining the inhibitory effect of F-actin on ART´s ADP-ribosylation activity.

Though CDTa, C2I and Iota-a toxin modify only monomeric but not filamentous actin (see also Figure 1A–C), this result does not necessarily indicate a negligible in vivo effect of Arp2 ADP-ribosylation. Only a small fraction of the total actin is in monomeric state in established cell lines, tissue epithelial or migrating white blood cells. Instead, most of the intracellular actin is polymerized to filamentous structures forming static or dynamic supramolecular organizations by interacting with other actin binding proteins, which may further inhibit ADP-ribosylation of actin as shown also for the TccC3 toxin [23]. It has been estimated that the intracellular concentration of actin is about 40- to 100-fold higher than that of Arp2/3 complex [28]. Our data suggest that actin and Arp2 were almost equally strongly modified in treated Caco2 cells and mouse colon. This observation will most probably be due to the fact that intracellularly only a small amount of the total actin is in monomeric state (5 to 10%), of which only a smaller fraction will not be complexed to G-actin binding proteins like thymosin ß4 or profilin, which might further reduce the accessibility of bacterial ARTs [23]. Additionally, capping of the plus-ends of F-actin by ADP-ribosylated actin [36] will reduce the rate of depolymerization of intracellular actin filaments, since then actin subunit dissociation occurs only slower from their minus ends.

In epithelial cells the F-actin is additionally stabilized by interactions with a large number of actin binding proteins. The Arp2/3 complex is of paramount importance for the connection of F-actin to for instance to adherence or tight junctions, which firmly attach neighboring cells to each other, or the stabilization of the cortical F-actin beneath the plasma membrane. Both structures appear to be affected by clostridial toxins possessing ART activity as shown by the effects of CDTa on the contacts of Caco2 cells and the dissolution of the epithelial layer of the mouse colon mucosa. The disruption of the cortical F-actin will reduce the resistance of the plasma membrane and lead to the observed outgrowth of microtubule bundles, a process that further increases pathogen attachment [18]. Infection of Caco2 cells by CDTa showed a translocation of Arp2 immunofluorescence away from the plasma membrane into the cell interior and the formation of large aggregates leading to disruption of the cell–cell and possibly also of cell-substratum contacts. This process will be aggravated by the simultaneous ADP-ribosylation of actin leading to the disassembly of the actin cytoskeleton [1,2,3,4]. The immunofluorescence data indicated 45 min after intoxication a partial colocalisation of Arp2 and actin in these aggregates. The preferential staining of F-actin by TRITC-phalloidin might, however, miss the presence of G-actin in these aggregates. Nevertheless, the intracellular formation of Arp2 and presumably also of Arp2/3 complex aggregations and the disassembly of the actin cytoskeleton will have arrested lamellipodial activity and intracellular vesicle transport terminating in cell death [4]. Our data suggest that an similar process occurs within the colon enterocytes after exposure to CDTa,b finally resulting in their detachment from the basal lamina.

Furthermore, the CDTa component might perform a dual toxic effect when inhibiting the function of Arp2/3 complex as it will also lead to an interruption of essential signaling pathways to the actin cytoskeleton affecting also processes like morphogenesis, cell motility, intracellular vesicle transport, and phagocytosis. In this respect, the action of toxins like CDT appears to have similar consequences as the lack of the Arp2/3 complex activating factor WASP in patients with Wiskott-Aldrich syndrome, which suffer from thrombocytopenia, an insufficiency of the immune cells to migrate and form contacts (as seen in CDTa-treated Caco2 cells), and gastrointestinal hemorrhage [6,7,8].

Finally, it has been shown that the Arp2/3 complex attaches via its additional subunits ArpC2, ArpC3 and ArpC4/or5 to the mother filament [27]. A recent cryo-electron study has shown that the main contacts of the Arp2/3 complex with the growing daughter filament are established by both Arp2 and Arp3 [39,40]. Therefore, it appears plausible that ADP-ribosylation of Arp2 will inhibit the addition of actin subunits and thereby the growth of a daughter filament. Furthermore, a recent study also showed that the additional ArpC2 and ArpC4 subunits form an extensive contact area with the mother filament [40]. Our data show that exactly these subunits were also ADP-ribosylated by particularly the C2I and Iota-a toxins (Figure 1). It is tempting to assume that their ADP-ribosylation (of ArpC2 and ArpC4) might inhibit the attachment of Arp2/3 complex to a mother filament [40] and thereby additionally contribute to the failure of the modified Arp2/3 complex to nucleate actin polymerization and branch formation. This aspect will, however, necessitate future investigations.

In summary, our findings show that the Arp2/3 complex is an additional target of toxins with ART-activity like CDT, C2I and Iota-a. ADP-ribosylation of Arp2 of the Arp2/3 complex inhibits its stimulatory activity on actin polymerization. Moreover, our studies employing culture cells and an intestinal tissue model suggest that the modification of the Arp2/3 complex together with that of actin induce complete disassembly of the actin cytoskeleton in Caco2 cells and most likely lead to the morphological alterations of colon tissue organization, especially of its epithelial layer that appear reminiscent of pseudomembranous colitis. These results may open new approaches and targets for the treatment for the severe diarrhea and pseudomembranous colitis caused by *C. difficile* and diseases induced by other clostridial bacteria.

## Figures and Tables

**Figure 1 cells-11-03661-f001:**
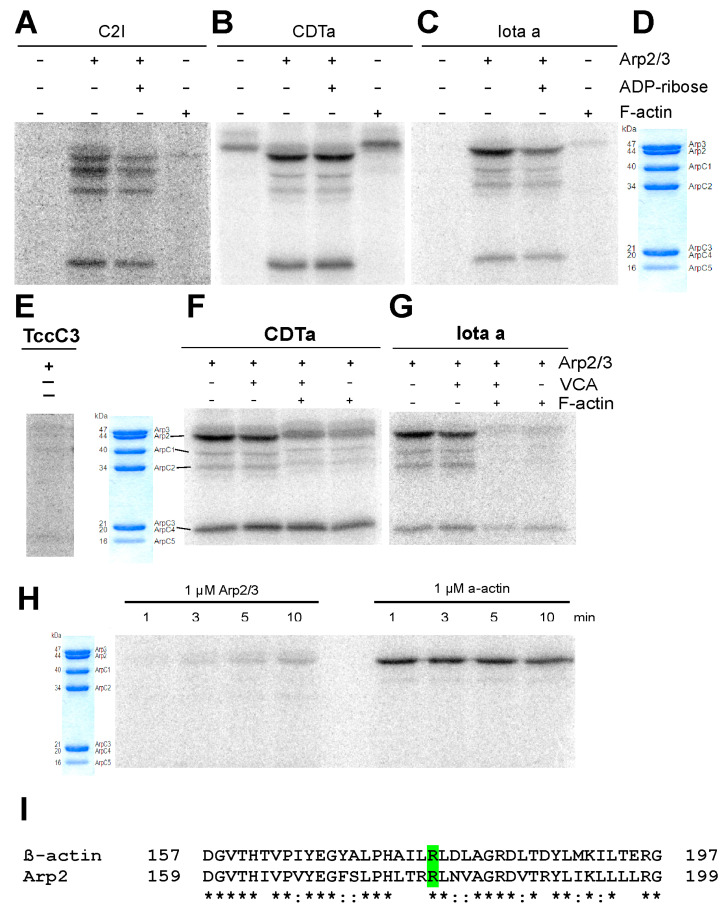
Autoradiographs using ^32^P-NAD to show ADP-ribosylation of Arp/3 complex components. by (**A**) CD1 (*C. botulinum*), (**B**) CDTa component (*C. difficile*), and (**C**) Iota-a toxin (*C. perfringens*). (**D**) Corresponding Coomassie blue stained gel (12.5% polyacrylamide) showing the components and their molecular mass of Arp2/3 complex purified from *A. castellani*. Note that Arp2 is most selectively ADP-ribosylated by CDTa and Iota-a toxin. Only the effect of Iota-a toxin is slightly inhibited by ADP-ribose. Only CDTa is auto-ADP-ribosyl^at^ed (4th lane in B). None of the toxins appears to ADP-ribosylate F-©in. (**E**) TccC3 (*Photorhabdus luminescence*) shows no ADP-ribosylation of any Arp2/3 complex component. (**F,G**) ADP-ribosylation by CDTa and Iota-a toxin is not modified by the presence of the VCA peptide but inhibited by F-actin. Incubation time in (**A**–**G**) was 90 min at 25 °C. (**H**) Time dependence of ADP-ribosylation of Arp2 of Arp2/3 complex and monomeric skeletal muscle G-actin by the CDTa component. (**I**) Gives the aligned sequences of *Acanthamoeba* and mammalian Arp2/3 complex around the sites of ADP-ribosylation (for details see text). (*) Identical residues and green highlighted are the ADP-ribosylated arginines.

**Figure 2 cells-11-03661-f002:**
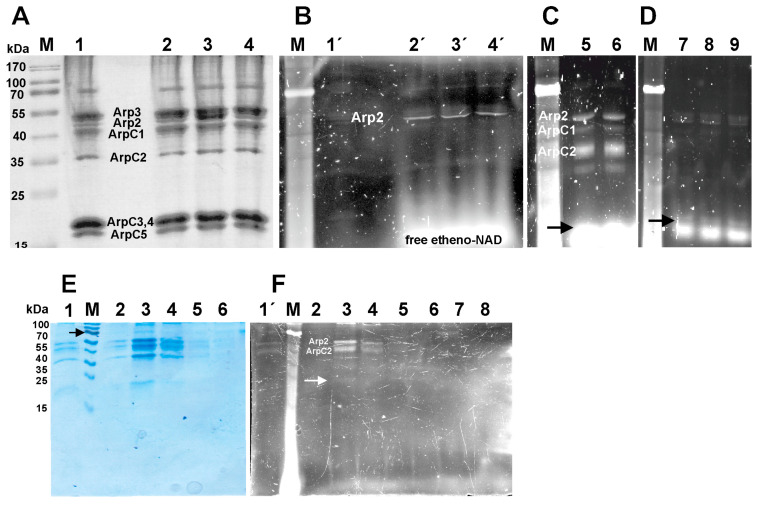
ADP-ribosylation of pig brain Arp2/3 complex by Clostridial toxins using etheno-NAD. Pig brain Arp2/3 complex (20 µL of 10 µM) was incubated without lane 1 or with 0.5 µg of CDTa, C2I, and Iota-a toxin (lane 2 to 4) in the presence of 0.4 mM etheno-NAD. After 2 h at room temperature the reactions were stopped with hot sample buffer and applied to a 15% polyacrylamide gel. The gel was analyzed for fluorescent bands by using a BioRad XR+ system using the “oriole” setting (**B**). Subsequently the gel was stained with Coomassie blue (**A**). The Arp2/3 complex subunits are indicated. Lanes 2´,3´, and 4´ give the fluorescently stained bands corresponding to mammalian Arp2. (**C**) Labelling of pig Arp2/3 by CDTa for 4 h under identical conditions: lane 5 without and lane 6 with 20 µM VCA-peptide. (**D**) Incubation of pig Arp2/3 with CDTa, C2I, and Iota-a toxin (lane 7 to 9) and 0.1 mM etheno-NAD for 2 h. (**E**,**F**) Separation of free etheno-NAD from pig Arp2/3 complex (50 µL of 10 µM) treated with CDTa for 5 h at 25 °C was subsequently separated from etheno-NAD by gel filtration using a Sephadex G75 column (1.0 × 5 cm). After applying 50 µL the treated Arp2/3 complex, the column was washed with HEPES-buffer and the following fractions were collected: 0.5 mL in the 1st fraction and thereafter 0.2 mL for fractions 2 to 8. The collected fractions were analyzed by 15% SDS-PAGE with a protein concentration of 2 µM for the fraction 2 to 4. (**E**) Gives the Coomassie-blue stained gel and (**F**) the corresponding fluorescent image obtained before staining. Lane 1 gives a small amount (about 10 µL) of labelled Arp2/3 complex before loading to the column and lanes 2–8 the fractions collected; Lane 1 corresponds to first fraction (0.5 mL), thereafter the 0.2 mL fractions. Black arrows in (**C**,**D**) and also white arrow in (**F**) point to weak labeling of ArpC4,5. Black arrow in (**E**) points to molecular mass marker of 70 kDa.

**Figure 3 cells-11-03661-f003:**
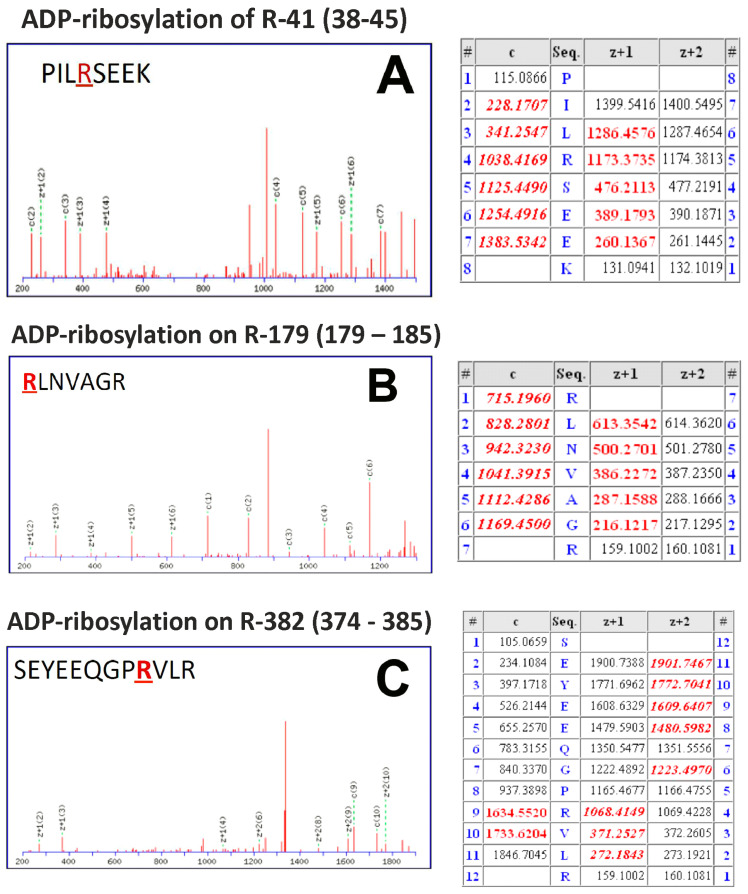
Mass spectrometric identification of ADP-ribosylation sites of Arp2. ADP-ribosylation sites were pinpointed from ETD spectra at positions R41, R179, and R382. Three peptides were identified: (**A**) ETD fragment ion mass spectrum of the peptide PILRSEEK (residues 179–185) with the ADP-ribosylated arginine at position 41; corresponding calculated fragment ion masses (c and z ions); values marked in red have been detected in the ETD spectrum. (**B**) ETD fragment ion mass spectrum RLNVAGR (residues 179–185) with the ADP-ribosylated arginine at position 179 and corresponding calculated fragment ion masses (c and z ions); values marked in red have been detected in the ETD spectrum. (**C**) ETD fragment ion mass spectrum of the peptide SEYEEQGPRVLR (residues 374–185) with the ADP-ribosylated arginine at position 382 and corresponding calculated fragment ion masses (c and z ions); values marked in red have been detected in the ETD spectrum. Peptide (**B**) containing arginine 179 was most prominently ADP-ribosylated.

**Figure 4 cells-11-03661-f004:**
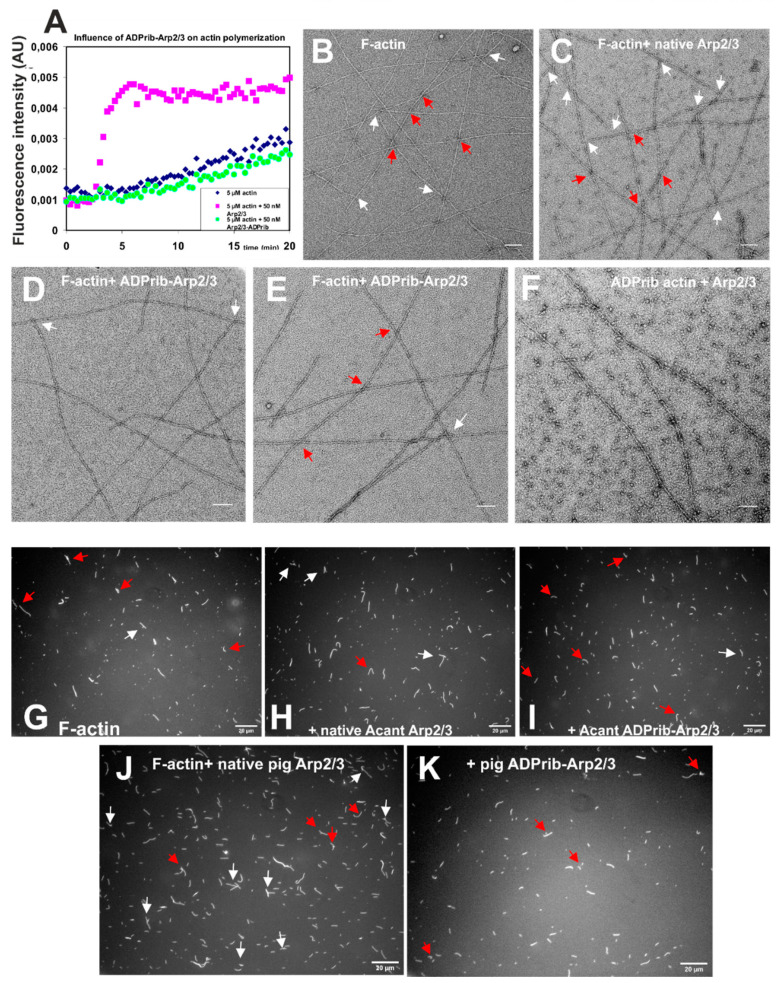
Properties of Arp2/3 complex after ADP-ribosylation of Arp2. (**A**) Pyrene-labelled actin (10%) was used to test the nucleating activity of native Arp2/3 before and after Arp2 ADP-ribosylation. Polymerization of 5 μM skeletal actin was initiated by addition of 2 mM MgCl_2_ in the absence or presence of 50 nM Arp2/3 complex. Note the absence of nucleating activity of Arp2/3 treated with His-tagged CDTa component for 4 h. The CDTa was removed by Ni-NTA-beads and further gel filtered to additionally remove its substrate NAD. Ordinate gives fluorescence intensity in arbitrary units (AU) ranging from 0 to 0.006; Abscissa gives time in min. (**B**–**F**) EM after negative staining of skeletal actin polymerized by 2 mM MgCl_2_ in the absence and presence of native and modified Arp2/3. (**B**) F-actin alone; (**C**) in the presence of native and (**D,E**) of CDTa treated Arp2/3 complex, and (**F**) ADP-ribosylated actin in the presence of native Arp2/3 complex. Branches are marked by black and crossings by red arrows (B-E). Magnification: 20.000 fold; bars correspond to 100 nm. (**G**–**K**) Fluorescent images of TRITC-phalloidin stained 10 nM actin samples: (**G**) F-actin control, (**H**) plus *Acant. castellani* native Arp2/3 complex, (**I**) plus *Acant. castellani* Arp2/3 complex after ADP-ribosylation; (**J**) plus pig brain native Arp2/3 complex, and (**K**) plus pig brain Arp2/3 complex after ADP-ribosylation (for details see text). Bars correspond to 20 µm; branches are marked by white and crossings by red arrows.

**Figure 5 cells-11-03661-f005:**
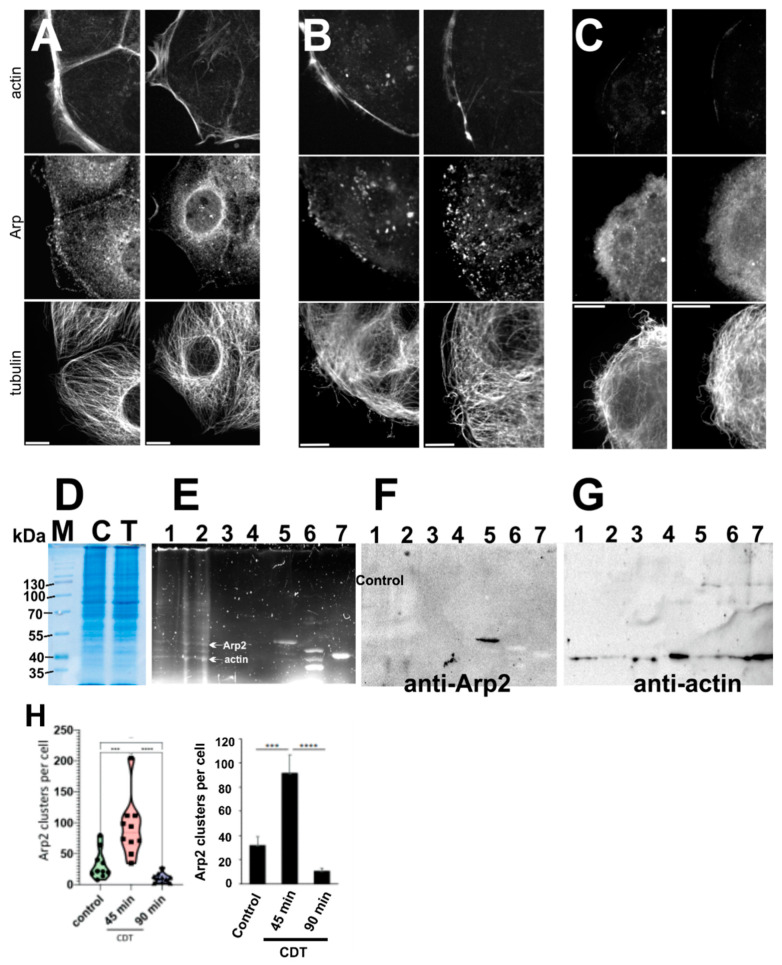
Effect of CDTa on Caco2 cells. (**A**–**C**) Caco2 cells seeded n coverslips were exposed to the CTDa and CDTb components to allow intracellular transport of CDTa by the membrane insertion CDTb component. (**A**) Control before intoxication, (**B**) 45 min and (**C**) 90 min after intoxication. After the indicated incubation periods, the cells were fixed and stained with TRITC-phalloidin (top rows). Anti-Arp2 mAb (middle rows); and anti-tubulin (lower rows) (for details see text). Bars correspond 10 µm. (**D**–**G**) Caco2 cells were incubated with CDTa and CDTb components, 0.1 mM etheno-NAD and 0.15 mM TritonX-100 to allow etheno-NAD diffusion into the cells. After 2 h or 3 h the cells were collected, washed 5 times in PBS and finally taken up in RIPA buffer to solubilize the cell content. After centrifugation the supernatants of the cell homogenates were analyzed by SDS-PAGE (12.5% polyacrylamide): (**D**) Coomassie blue staining of the control (**C**) and treated (T) cells. (**E**) Gives SDS-PAGE of cell homogenates and of the anti-Arp2 immune-precipitation (IP); the protein bands were fluorescently stained by inclusion of 0.5% trichloroethanol into the gel and visualized by the “oriol” setting of the BioRad transilluminator. (**F**) Gives the immunoblot of gel shown in (**E**) with anti-Arp2 mAB; and (**G**) with anti actin. Lanes in (**E**–**G**): Lane 1: homogenates of control and lane 2 of CDTa treated cells. Lanes 3 and 4: washing steps before final IP; lane 5: final anti-Arp2 IP showing Protein-A beads bound fluorescent material; lane 6: etheno-NAD labelled *Acanthamoeba* Arp2/3 complex; and lane 7: etheno-NAD labeled actin. Note lower molecular mass of Arp2 of *Acanthamoeba* Arp2/3 complex (upper band of lane 6). (**H**) Statistical evaluation of Z-stacks of cells treated and stained as in (**A**–**C**) were 3D-reconstructed (z-Step: 0.4 µm) with Metamorph and the number of Arp2 accumulations with 0.07 to 1.7 µm^3^ was quantified per cell (ordinate) for the control and the cells treated with the CDTa and CDTb toxins (CDT) for 45 and 90 min. Differences were statistically analyzed by 1-way ANOVA with Tukey post-test. 10 fields of view were analyzed for each condition with ≥34 cells in total, from 2 independent experiments. The data are represented both in the “violin” mode and as columns. Asterisks indicate significant differences (*p* ≤ 0.001:***; *p* ≤ 0.0001: ****).

**Figure 6 cells-11-03661-f006:**
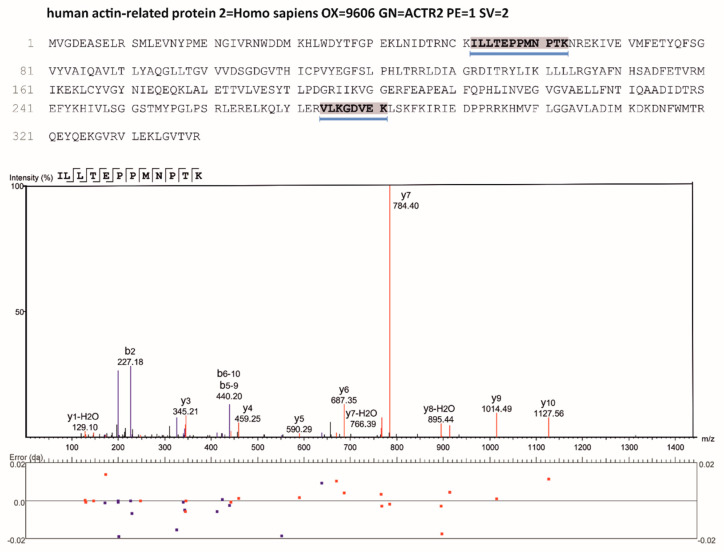
Mass spectrometric identification of human Arp2 unique peptides. Fluorescently labelled band after IP as shown in lane 5 of Figure 5G was excised from a similar SDS-PAGE gel and digested with trypsin as detailed in Materials and Methods. Two tryptic peptides (residues 52 till 63 and 274 till 281) unique to human Arp2 (gene: ACTR2) were identified by nanoLC-MS/MS (highlighted in the protein sequence). The lower part shows the fragment ion spectrum (HCD) of one of the identified tryptic peptides demonstrating high peptide sequence coverage and thus reliable identification of Arp2.

**Figure 7 cells-11-03661-f007:**
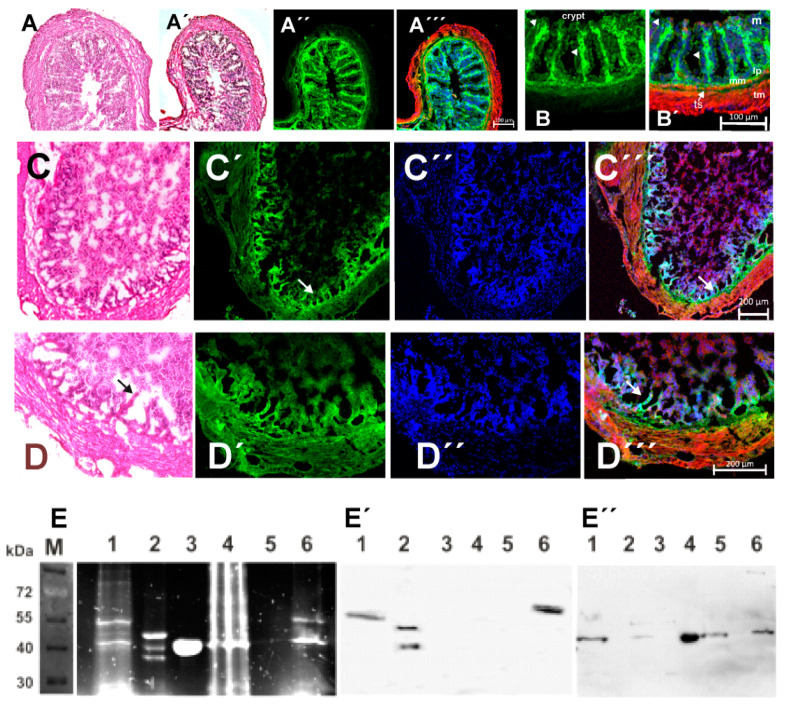
Effect of *C. difficile* CDTa and CDTb toxins on the mucosa of mouse colon. Mouse colon pieces filled with PBS alone (**A**) or plus 0.15 mM Triton X-100 (control) (**A´**) were ligated at both ends and incubated for 3 h at 37 °C. (**A**,**A´**) Hematoxylin stained; note no gross alterations in the organization of the mucosa. (**A´´**,**B**) Consecutive sections stained with anti-Arp2 monoclonal antibody and (**A´´´**,**B´)** merged images with TRITC-phalloidin staining. Arrowheads point to weak anti-Arp2 staining of the enterocytes, in (**B´**) the intestinal layers are marked as detailed in text. (**C**,**D**). Sections of colon pieces were incubated for 3 h with PBS, 0.15 mM Triton X-100, the CDTa and CDTb toxin components both at 0.1 mg/mL, and 0.1 mM etheno-NAD. Triton X-100 was included to secure diffusion of etheno-NAD into the mucosal enterocytes. (**C**,**D**) Cryo-sections after 3 h incubation: (**C**,**D**) H.E. stained; (**C´**,**D´**) immunostained with anti-Arp2; (**C´´**,**D´´**) Hoechst 33342, and (**C´´´**,**D´´´**) merged images together with TRITC-phalloidin stain. Arrows point to anti-Arp2 positive remnants of the Lamina propria within of the crypts, whereas the enterocytes appear completely dissolved. For details see text. Bars in (**A´´´**,**B´**) represent 100 μm and in (**C**,**D**) 200 μm. (**E**) SDS-PAGE (12.5% polyacrylamide) of IP fractions of colon homogenates after exposure for 3 h to the CDTa and CDTb (*C. difficile*) components and 0.1 mM etheno-NAD analyzed for fluorescence, and Western blots immunostained with anti-Arp2 (**E´**), and anti-actin (**E´´**). Lane 1: final IP fraction of colon segments treated for 2 h, lane 2 and 3: purified *Acanthamoeba* Arp2/3 complex and actin, both fluorescently labeled covalently with etheno-ADP. Lane 4: Colon homogenate after 3 h treatment, lane 5: supernatant of final IP washing and lane 6: precipitate of final IP washing containing the protein A beads.

**Table 1 cells-11-03661-t001:** Statistical evaluation of F-actin branching events visualized by EM or after TRITC-phalloidin by fluorescence microscopy (LM). (A) Non-overlapping electron microscopical images of skeletal muscle F-actin alone or after incubation with *Acamthamoeba* Arp2/3 complex before or after ADP-ribosylation (27 images of each condition) were visually evaluated for branching events. The number of filaments and branching events were manually counted for all images (see also Figure 4B–E). (B) Determination of branching events after staining with TRITC-phalloidin of F-actin alone or plus *Acanthamoeba* or porcine brain Arp2/3 complex before and after ADP-ribosylation. The number of total filaments and the branching events in each selected image were determined by ImageJ ridge detection (Figure 4G–K). Details of the procedure of filament counting and their statistical evaluation by ImageJ are given in the Material and Methods. The branching events are given as percentages of the total filaments or as events occurring for each single filament (giving also the standard error of the mean; SEM). Note the reduction of branching events in the presence of ADP-ribosylated Arp2/3 complex to the value of F-actin alone. The percental difference in branching events observed between EM and FM are probably due to the difference in the concentrations of the F-actins: For EM the F-actin subunit concentration was 2.35 µM and for fluorescence microscopy (FM) it was reduced to 10 nM, because at higher concentrations the filaments were too numerous and crowded to distinguish between crossing and branching.

Sample	Images	Total Filaments	Branching Events	Branching/Total Filaments *	Av.Branching/Filaments ** ±SEM
**A—Electron microscopy.**					
F-actin	27	475	35	7.37%	7.34 ± 1.23%
+Acant Arp2/3	27	524	67	12.79%	13.49 ± 1.78%
+Acant Arp2/3 ADPrib	27	304	30	9.87%	10.50 ± 1.28%
**B—Fluorescence microscopy.**					
F-actin	46	6374	28	0.44%	0.39 ± 0.08%
+Acant. Arp2/3	30	6672	76	1.14%	1.13 ± 0.12%
+Acant. Arp2/3 ADPrib	20	4198	21	0.50%	0.47 ± 0.08%
+pig brain Arp2/3	85	11857	90	0.76%	0.77 ± 0.09%
+pig brain Arp2/3 ADPrib	70	7348	24	0.33%	0.30 ± 0.06%

* Total branching events/total filaments. ** Average value of branching/filaments in each image.

## Data Availability

Data and materials are available from the authors.

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
