# Peer review of "Inhibition of Arp2/3 Complex after ADP-Ribosylation of Arp2 by Binary Clostridioides Toxins"

_cells, 2022, doi:10.3390/cells11223661_

Round 1
Reviewer 1 Report
The manuscript was well written, and the experiments were well performed. Possibly, it would be better to add the experiments to evaluate the cell motility of bacterial toxin-treated cells such as wound healing assay or boyden chamber experiments, since Arp2/3 is well known to be related with cell migration/motility.
Author Response
Answer to reviewer 1
It is certainly a very interesting proposal to test the effects of the toxins on migrating cells.
We have chosen the Caco2 cells and the staining procedure because of the following reasons:
- We wanted to use a cell line, which is close to colon enterocytes to ensure that they have the CDTb receptor (the lipolysis-stimulated lipoprotein receptor) so that they can take up the clostridial toxins by their “natural” mechanism. We assume that for achieving cellular uptake of the toxins it would have necessitated much harsher methods than necessary for the uptake of etheno-NAD.
- Such an experiment would also lead to ADP-ribosylation actin and subsequently to loss of cell-cell contacts (as we have described) and also to detachment of the cells from the substratum.
- Our data suggest that toxin uptake leads to cell death (also reviewed by Kordus et al. Nat Rev Microbiol 2022). Therefore, inhibition of cell migration by Arp2/3 complex containing ADP-ribosylated Arp2 or ensuing cell death might be difficult to separate.
Alternatively, one could try to selectively microinject into cells Arp2/3 complex with ADP-ribosylated Arp2. This however will be a separate project, worth thinking about anyway. But too much for the moment.
Reviewer 2 Report
The manuscript deals with the effects of the clostridial toxins CDT, C2I, and Iota-a on the actin polymerization by acting on the Arp2-3 complex.
In my opinion, the study design is accurate, the obtained data convincing and the conclusions well supported.
However, the paper needs an important formal revision. In particular, the sections need to be reorganized, as some experimental procedures are missing or described in the Results Section, result argumentations are similarly provided in the Results Section, while the Discussion Section remains a sterile description and recapitulation of previous data.
The manuscript needs some linguistic revision since the language is quite accurate but several punctuation or syntax issues have to be fixed.
Following there are specific comments.
Lines 17-21: Please revise the punctuation of this sentence.
Line 31: Please use “infectious” instead of “infection”.
Line 32: This sentence is syntactically incorrect (their proteins…of these pathogens).
Lines 34-35: The scientific name of bacterial species should be provided in full at their first occurrence.
Line 41 and elsewhere in the text: Please add a hyphenation (ADP-containing). Since a fair amount of linguistic errors have to be fixed, no more language comments will be provided. The manuscript must be carefully revised.
Line 43: please specify the protein to which Arg177 belongs.
Line 48: please provide a reference for this statement.
Line 77: This is the first occurrence in the text for Escherichia coli, and its scientific name should be provided in full.
Lines 81 and 84: Reference #17 does not deal with the preparation of clostridial toxins.
Lines 96-101: Please provide full details of the authorization for the experimental use of animals.
Line 101: The experimental procedure should be produced in the M&M Section, not in the Results Section.
Line 159: Please check the URL.
Line 177: Please provide details about the protease.
Lines 250-255: Neither details about experiments with formins, nor with TccC3 have been provided, please add them in the material and methods section.
Lines 270-271: The description of WASP-VCA should be provided in full at its first occurrence, at line 76.
Lines 277-284, and elsewhere in the text: The Results Section should only be devoted to presenting results, without discussing them: That should be made in the Discussion section, unless the two sections were merged.
Figure #5 may be split for improving readability.
Line 488: Only immunostaining was described in the M&M section. Please add details for the immunoprecipitation procedures.
Line 527: Please do not start the section with “finally”. Similar to what?
Lines 527 and following: please see the comments to line 101.
Line 548: Maybe the Authors intended 7B’?
Lines 618-637: This long part is completely dedicated to describing the role and mechanisms of actin polymerization, with few references to the clostridial toxin role according to the gathered data. Please also see the general comments.
Lines 642-646: What could be the biological consequences of such an effect?
Lines 658-659: It is not clear how the cited reference may support the proposed hypothesis.
Lines 667-670: This aspect, along with many others, has been poorly discussed.
Line 671: Please abbreviate the genus name, as it is not its first occurrence.
Author Response
Answer to reviewer 2
I have amended the manuscript according to all points raised by reviewer 2.
In particular I have added
- Full details about the authorization for the use of the animals. Since in our case the animals were used only for organ excision, we do not need a specific experimental approval, but we have to comply with the Animal Care Regulation.
- I have included the details of the purifications of TccC3 and the formins.
- I have added a description of the immune-precipitation procedure in the M&M section.
- I have extended within the Discussion the paragraph described the Arp2 aggregates and their possible consequences together with the effect of the disassembly of the actin cytoskeleton.
- All alteration in the resubmitted manuscript are marked in red.
Reviewer 3 Report
This manuscript investigated how clostridioides bacteria binary toxins are able to ADP-ribosylate Arp2/3 complex components : Arp2, ArpC1, ArpC2 and Arp4/5. The Arp2/3 complex has been shown to be a target for clostridial ADP-ribosyltransferase remodeling. However, the mechanism by which this pathogenic mechanism occurs is not well understood. This study aimed to analyze the ART-activity containing toxins due to their ability to ADP-ribosylate. Arp2/3 complex by multiple approaches including EM imaging, fluorescence imaging, Mass spectrometry, and cell and tissue modeling studies. Their results show that the ADP-ribosylation of Arp2 of Arp2/3 complex inhibits its stimulatory activity on actin polymerization. This reviewer suggests the authors considering the following comments in their revised manuscript:
1. The main strength of this manuscript is that the authors employed multiple complimentary approaches, ranging from direct visualization of branches to the effects on tissues, in their study. On the other hand, because of this fact, the conciseness of the manuscript seems to get sacrificed. It would be great if the authors could improve delivering their main messages more clearly by clearly and more concisely stating their points.
2. The resolutions and/or quality of some figures need to be improved. For example, Fig. 3 and Fig. 5H are too blurry to read. Put legible axis labels (y-axis as fluorescence intensity, and x-axis as Time) for Fig. 4A.
3. Can the authors explain the discrepancy between % branches in EM and fluorescence microscopy images (Table 1)?
Author Response
Reply to reviewer 3
- I have tried to give a general description of our idea about the composition of the manuscript, i.e. from biochemical data to the tissue showing the possible pathological effects caused by the presumed protein modifications.
- I have added the description of the axis of Fig. 4A and 5H.
- It was however nor possible to appreciably improve Fig.3
- I give an explanation about the different percental values of branching in the legend to Table 1.
Round 2
Reviewer 1 Report
Current version of manuscript can be accepted for publication in Cells.